# Dogs at the Workplace: A Multiple Case Study

**DOI:** 10.3390/ani11010089

**Published:** 2021-01-05

**Authors:** Elisa Wagner, Miguel Pina e Cunha

**Affiliations:** Nova School of Business and Economics, Universidade Nova de Lisboa, 1099-085 Lisboa, Portugal; wagner.elisa92@googlemail.com

**Keywords:** human–dog relationship, dogs in organizations, pet-friendly policies

## Abstract

**Simple Summary:**

Pet-friendly policies are becoming more common in the workplace, but little is known about how the presence of dogs influences the work environment. For this reason, the paper includes a study on how dogs influence the work environment and under which circumstances companies can benefit from a pet-friendly policy. An inductive research approach was used for this study. Qualitative data were conducted by interviewing dog-owners who routinely bring their dogs to the office and persons in management positions in the same companies. Finally, the results of the evolving data analysis through successive iterations was formed into a grounded theory. The results of the study generally support the belief that dogs at the workplace can have a positive influence on individual and collective well-being of organizational members in an office environment. However, the study shows that these positive effects of dogs are linked to certain prerequisites that need to be fulfilled in the company. These include flexible hours, autonomy, and open communication. While those requisites are not necessary in order to be able to implement pet-friendly policies, they are highly relevant when companies seek to decrease stress for employees and improve the work atmosphere and social capital.

**Abstract:**

As the work environment is increasing in competitiveness and stressfulness, more and more companies try to increase employee well-being. One option is allowing employees to bring their dogs to work, building on the considerable evidence that dogs have a positive influence on people’s well-being. However, little is known about how a dog’s presence influences the employees and the companies in offices. Therefore, we empirically scrutinize the presence of dogs in organizations and the impact of pet-friendly organizational policies, with multiple case studies with semi-structured interviews as their foundation. Based on an inductive approach for the data analysis, we found that organizational members consider that dogs can lower their stress, improve communication, and foster social cohesion when a flexible organizational culture is in place. This includes the following: Problems in the company are openly addressed; employees have job autonomy, with flexibility to take breaks; and mistakes and errors are allowed to be made by employees and their companions alike, and room to find solutions is given. The inflexible permission of pets at work can, on the contrary, create pressure and stress in employees. For the business world, this implicates that this kind of incentive only leads to success if the right framework and culture is in place, and it cannot only be seen as an instrument to increase employee well-being.

## 1. Introduction

An emerging but still controversial topic in management and organization studies is the pet-friendly workplace. The fact that companies such as Google, Apple, and Amazon already allow dogs at the office [1] has provided some debate [1,2,3] but failed until now to stimulate systematic empirical research on the topics of pets at the workplace [4,5].

Little is known, therefore, about how the presence of dogs is experienced in actual work environments and what the prerequisites are for companies to be able to benefit from pet-friendly policies, without incurring undesired costs. For this reason, we investigated how organizational members perceive the way(s) that dogs influence the work environment and under what circumstances companies can benefit from pet-friendly policies.

To answer these questions, an inductive research approach was adopted. We conducted a study based on qualitative data with dog-owners who routinely bring their dogs to the office in the companies where they work. Finally, the results of the evolving data analysis through successive iterations was formed into a grounded theory. We were interested in taking a more detailed look at interpretations, to provide an actor-sensitive understanding of the phenomenon [6]. Overall, we contribute by offering an empirically nuanced view of the presence of dogs in organizations, finding that pet-friendly policies can positively impact the workplace, but only when it reflects a genuine approach in line with a company’s values and its culture, rather than a merely instrumental one only used to create an impression.

## 2. Theoretical Background

### 2.1. Dogs and Humans—In General

Human–pet interaction and bonding, especially with dogs and cats, is an interspecies relationship having roots in pre-history [7]. Dogs have been labeled the “best friend” of humans, highlighting the depth of the relationship.

The therapeutic value of dogs has been the focus of various studies. The general belief holds that dogs positively impact people who enter into contact with them; however, some studies also show no or a negative effect on human well-being [8,9,10]. Dog-accompanied therapy has been proven to be quite effective [11,12,13], especially with children, the vulnerable, and the elderly [14]. According to published research, dog-owners benefit from this relationship by obtaining a boost in mental and physical health. Studies also relate dog-ownership to reduced depression [15,16], and especially the elderly benefit from dogs as companions [17]. One reason why psychological well-being seems to increase with pet interaction is the function of social support that they bring. According to Foreman, Glenn, et al. [1], “the term ‘social support’ is often used to describe the mechanisms by which relationships with other people buffer individuals from stress” due to a social bond. Scientists found an oxytocin positive feedback loop between dogs and humans [18], a hormone that plays an important role in the development of human bonds, leading to the belief that the bond between dog and human can also act as social support for humans. Further, studies show that dogs offer a certain kind of social support for owners and that the animal relationship diminishes negative emotions resulting from rejection experiences [19]. Participants in the cited study report less depression, less loneliness, and a greater subjective happiness. In addition, dogs make a unique contribution to the well-being of their owners beyond simply substituting for peer social support [19].

Another factor in the role that dogs play in boosting humans’ mental well-being is the relaxing influence of dogs on humans. The influence of dogs has been linked to stress release in several studies. Reduced levels of cortisol, a stress hormone, have been detected in positive dog–human interaction [18]. The presence of a companion dog reduces the stress levels of women in a way similar to talking to a close friend [20], and the presence of unfamiliar dogs decreases cortisol levels and heart rate as physical indicators for lower stress [21,22]. Miller et al. [23] found that women might experience the relaxing influence of dogs more than men do. Other studies by Vormbrock and Grossberg [8], Straatman et al. [9], and Gee et al. [10] did not find evidence of a link between human stress and dogs—neither in a change in blood pressure nor in stress test results. While some researchers believe that the difference in findings can be explained, to some degree, by the familiarity of the dog and the environment with familiar dogs having a more positive influence on the owner’s stress level than unfamiliar but friendly dogs [9,21], the data are not conclusive. While those studies cannot answer the question of whether dogs really reduce physical stress indicators, owners and people interacting with dogs certainly believe that dogs help decrease stress [24]. Studies measuring self-reported stress show a significant effect of dogs on stress when interacting with one’s own or an unfamiliar dog [25]. Self-evaluations reports of greater stress have been connected to lower levels of well-being in multiple studies [26,27,28], and indicated that dogs have a beneficial effect on the stress level and psychological well-being of humans, whether physically measurable or not.

The impact of dogs on humans’ physical well-being is nevertheless mixed and not consistent through the literature. While some studies relate owning a dog to health benefits, such as lower blood pressure [15,29] and faster recovery from illnesses [16,30], other studies report even negative impacts on the health of dog-owners in regard to owning a dog [31]. However, studies indicate that dogs have a positive impact on the amount of exercise of their owners [15,32,33], which is widely agreed to be related to better health.

In addition, dogs can also have positive influences on group dynamics, social interactions, and social behaviors [34,35], and trust is rated higher in groups in which a dog is present [34]. Dogs tend to improve the quality of social interactions in group therapy settings [35], with strangers and also with friends and acquaintances, which even leads to friendships and more social support in groups, thereby influencing long-term relationships in communities [36,37,38]. This might be related to a dog-related increase of oxytocin in humans, which is linked to an increase of pro-social behavior [39].

### 2.2. Dogs in the Workplace

In light of the considerable evidence for a positive influence of dogs on people’s well-being, it is little surprise that some companies implement pet-friendly policies. This is especially true considering that employees operate today in competitive and stressful environments and that employers realize that dogs can promote employee well-being. Pet-friendly policies can be defined as “rules, guidelines, and procedures that accept, welcome and regulate the presence of pets into the working environment, in order to benefit from the human-animal bond and interaction” [38]. Tech companies in Silicon Valley, including Amazon, Google, and Apple, have had pet-friendly policies in place for years. At Amazon, around 2000 dogs are regularly brought to work at the main campus [40]. In general, people bring dogs more often when working in smaller offices [38], in smaller and more creative companies, and in non-profit institutions [41].

Burnout tendencies, which are related to higher stress and dissatisfaction, increase the likelihood of turnover intention [42]. Finding ways to reduce stress for employees is therefore important. The work environment can benefit from the relaxing influence of dogs, reducing stress for owners and co-workers, but the evidence is mixed. A recent study found that while there is no difference in cortisol level between dog-owners and non-dog-owners, people who brought their pet companions to work have lower self-reported stress at work [43]. In addition, the highest perceived function of pets in the workplace is to reduce stress, reported by owners, managers, and non-owners [44]. Allowing dogs into the workplace can also reduce the stress related to the personal life of dog-owners. Dog-owners who are allowed to bring a dog to work feel less stressed than do colleagues who either leave their dog at home or in a dog-care facility [45], making pet-friendly policies an instrument for better employee work–life balance.

While all of these factors might relate to job satisfaction, engagement, and turnover intentions, surprisingly, little research has addressed the topic. Hall and Mills [38] found significantly higher self-reported work engagement and lower turnover intention when pet-friendly policies are in place. This takes further the findings of Barker [46], who reported that employees who are allowed to bring their pets to work scored higher on multiple job satisfaction scales. Moreover, people who bring their dogs to work rate their companies higher on benefits and organizational support than employees who do not own or bring dogs to work [38,43].

#### 2.2.1. Interaction and Group Dynamics

Open communication and trust are seen as highly important company characteristics for employees [47], and relationships with colleagues are one of the most important factors for employee job satisfaction [48]. Therefore, creating a positive social environment for employees should be a high priority for companies. With dogs having positive impacts on human interactions, these benefits could be transferred to the office as well. Some managers have taken notice that the influence of dogs on the social capital of a community is transferable to the workplace as well [5]. Nevertheless, little is known about how dogs influence the community in the workplace. To the best of our knowledge, only one office-related investigation has been performed in this regard: Hall and Mills [38] found that people who bring their dog to work often scored above average on friendship acuity within the office, supporting the idea that dogs can enhance social interaction and social bonding in a workplace setting.

#### 2.2.2. Employer Brand

People and places, in general, appear to be happier and more relaxed when a dog is present [49]. In addition, pet-friendly policies can project images of a relaxed culture and of informality, conveying a positive image to the outside world. Coming across as being dog-friendly is good press for companies [50]. It also attracts a certain kind of applicant, creating a benefit in the competitive job market [41]. This becomes even more relevant when taking into account that millennials will overtake the baby boomers soon as the largest pet-owning generation [40], and this generation is more likely to switch jobs rather than work for a low-reputation employer [51]. With this changing environment and expectations from employees, companies need to offer packages of benefits and policies that shape the workplace experience so that it relates to the needs of prospective and current employees [52]. For example, companies that have policies in place that support a work–life balance are more attractive to the young workforce [53,54]. Therefore, pet-friendly policies become more important for this new generation of workers.

Allowing dogs into the office also symbolizes a sense of values that applicants, as well as employees, can (or cannot) identify with [43]. According to social identity theory, people derive their self-concept from their membership in certain social groups and therefore join companies that fit their own values and are staffed with people similar to themselves [55]. Pet-friendly policies might help employees and applicants who are dog-owners, dog-friendly, or are more relaxed in general about rules and regulations to identify with a company. Seeing a dog in the office or a dog’s place at a desk (as seen in Figure 1) might has an impact on the applicant’s perception of the company. Consequently, pet-friendly policies can be an important component of the decision for or against applying for a specific job and have a role in employer branding [40,41].

### 2.3. Risks of Having Dogs at the Workplace

While the abovementioned benefits make pet-friendly policies attractive, certain risks cannot be neglected. The most obvious risks have to do with allergies, phobias, and personal dislikes, but also the potential of distraction coming from dogs. 

#### 2.3.1. Allergies, Phobias, and Dislikes

According to Allergy Asthma Immunol Research, an estimated 10–20% of people worldwide are allergic to dog hair [56], with symptoms varying in intensity and nature, from swollen and itching eyes and nose, to respiratory problems and body rashes. The threat of allergic reactions amongst co-workers or visitors toward dogs in the workplace is thus an important concern [1]. Companies also have to consider employees’ genuine phobias, lesser fears, and overall dislike of dogs. According to a Gallup poll, 11% of Americans are afraid of dogs [57]. Bringing dogs to work can increase stress and compromise the well-being and the feeling of safety of co-workers who are genuinely afraid of animals [2]. Some companies report that they disqualify applicants who dislike dogs or who have allergies [41]. In the competition for talent, it is also questionable if adding another filter to the recruitment process might lead to fewer applicants for jobs and therefore harm a company in the long run [2].

#### 2.3.2. Dangers

Companies may be afraid of dog bites and other injuries when considering implementing pet-friendly policies. According to an investigation in the UK on dog related musculoskeletal injury [58], dog bites are very rare. Most human injures involving dogs occur from tripping over the dog or its toys or getting pulled off one’s feet by a dog on a leash [58]. Consequently, companies need to consider legal responsibility. This might differ from country to country, but injured co-workers might have a justifiable claim of negligence against an employer for dog-related injuries that occur at the work site [1].

#### 2.3.3. Distractions

Another point that needs to be taken into consideration is the influence of dogs on productivity of those in the office. Dogs can be an extra responsibility for the owner, who needs to continuously keep the dog in mind. Some companies report problems with dogs stealing food, barking, and behaving aggressively toward other dogs in the office, which forces owners to intervene and neglect their work assignments [50]. According to Barker et al. [43], 20% of employees without dogs perceived dogs as reducing their personal productivity. In addition, dogs need to take breaks, go outside, and be fed during the work time. Employees need to take time off in order to take care of the dog—time that could be spent bonding with colleagues [2,41].

### 2.4. Limitations of Existing Research

While published research provides a good overview of how dogs influence humans in their personal life, little is known about the effects of dogs in organizational settings. Most theories regarding pet-friendly policies and the influence on the work community have started with the assumption that dogs’ influence in the office is similar to an outside setting, but studies pertaining to an office setting are rare. Therefore, the research question for this study is, how do dogs influence the work environment, and under what circumstances can companies benefit from a pet-friendly policy?

## 3. Methods

A multiple case study with five companies was conducted to examine how dogs influence organizations when they can be brought to the office. Inductive research using the grounded theory approach was adopted, giving voice to the interpretation of events by the people who actually experienced those events, making the insiders’ point of view the main foundation for the findings. Inductive research is particularly useful when exploring topics that are difficult to identify or measure, as is the case with the impact of dogs on organizations and the job environment [59]. It allows us to build our understanding of the properly contextualized experiences of those involved in companies with dogs, rather than imposing a particular framework upon them [60].

The five companies included in our multiple case study are from the creative agency sector, as this is a sector at the vanguard of pet policies, all located in a single country, (Germany) in order to minimize cultural differences. The companies varied in size from 6 to 40 employees. A detailed company profile is in Table 1.

One of the authors had already been in contact with the companies Bulldog and Labrador (all names are pseudonyms) in the past and therefore knew that they have dogs in their offices. The company Bulldog recommended Poodle as an additional interview candidate. The remaining companies, German Shepherd and Beagle, were identified for this research through their websites, where they showed pictures of dogs in the offices, making them suitable subjects for the research. Companies were contacted directly via mail or phone and asked if dogs are allowed and present and if they would be willing to participate in a case study.

Interviews are an efficient way to gather rich, empirical data, especially when focusing on phenomena that are episodic and infrequent [61]. We decided to interview a minimum of two persons per company and complemented the interviews with company data available in formal documents and websites, in order to increase our sensitivity about the different organizations. A total of 12 persons from the five focal organizations were interviewed in August and September of 2019. The interviewees were employees with dogs, managers without dogs, and managers with dogs at the workplace, thereby looking at the phenomenon of dogs at the workplace from diverse perspectives [61]. All informants were highly knowledgeable about the topic of dogs at the workplace and highly involved in the processes. The interviews lasted between 25 and 45 min, and at least one of each interview-set was conducted directly at the company. All interviews were held in German, then transcribed, and then translated to English. 

The interviews included questions regarding the policies regarding dogs, positive and negative effects noticed, and changes in work behavior when dogs are present, and managers were asked about their motivations for allowing dogs in the office. The interviews were semi-structured with a basic interview guideline (the interview guidelines are shown in Figure 2 and Figure 3) for both parties that was prepared beforehand and followed during the interviews. Questions were modified and added during the interviews to adapt and to find deeper connections. In addition to interviews, the first author engaged in direct observation of social interactions and the work environment (e.g., interaction of co-workers with the dog, physical setup of the company’s office facilities, and dog’s general sleeping place in the office), to uncover insights regarding the general feeling about dogs at the office and the environment of the companies. Following the interviews, the websites and social media accounts of the companies were analyzed on whether the dogs are made visible for outsides and at what frequency they are shown.

During the interviews, from one company to the next, more and more information and insights became repetitive, until at the last company (Beagle), no additional data were added. In that interview, no more first-order categories were added to the analysis, but only supported points already identified. According to Grossoehme [62] and Morse [63], the categories, and thereby the data, can be considered saturated when no new information is found during interviews. This led us to conclude that conceptual saturation had been obtained after our 12th interview, and no further interviews were performed. 

## 4. Data Analysis

The analysis followed established techniques and procedures for grounded theory building after Gioia (2013) and consisted of a series of steps to move from the data to theory [60]. The interview transcripts were analyzed for important terms and observations mentioned. Quotes were collected, mostly replicating informants’ terms and language. Quotes regarding the same topic from employees and managers were clustered and compared. From that, a total of 38 first-order categories were identified, with every category describing a similar phenomenon in multiple interviews and companies. The first-order concepts revealed key elements but no deeper patterns or relationship in the data. Table 2 illustrates examples of quotes for all of the first-order categories.

The categories were then examined and combined according to the most important key terms. For example, many people mentioned that taking breaks whenever necessary is important for having a dog in the office to deal with the animal’s needs, which was identified as a need for autonomy in job design. Only at this point was the literature taken under consideration more closely, comparing data with existing theories. Importantly, not only terms were taken into account that were mentioned by most or all participants, but also terms and connections that have not been mentioned in the literature as much, such as social cohesion resulting from the presence of dogs [62]. When an unknown phenomenon was found, the quotes of other interviews were scanned for information that supported or contradicted the new idea. These are second-order categories, which can be explained by the combination of first-order terms, all backed up with multiple quotes from the interviews and facts from observation. In this research, a total of 12 categories emerged.

In the third step of the analysis, the 12 major categories were brought together into four unique aggregate dimensions. For this, the relations between first-order themes and second-order categories were examined, and the overarching concepts regarding the influence of dogs in companies were captured. Further texts from the literature regarding employee satisfaction, culture, and policies were consulted, to compose independent but relevant dimensions. For example, the trial-and-error mentality, autonomy in job design, and open communication all point in the direction of a flexible organizational culture in which employees with dogs felt able to bring the dog and deal with the extra burden thus entailed. It became evident that these points were prerequisites to benefit from dogs in the office. From this analysis, the data structure, as shown in Figure 4, was formed.

## 5. Findings

Our data revealed that dogs constitute an added responsibility not just in the work environment. Interviewees often compared dogs with having a child, regarding the extra responsibility of taking care of a helpless dependent. The question of the whereabouts of the dogs during working hours is stressful for owners. In a work environment, the dogs need a fair amount of attention. Small disturbances like barking or playing are not unusual but are not seen as too much of a disturbance. Dog-owners also reported that a certain amount of attention is almost always given to the dog and that (it was said by the owners and the managers alike) this does not compromise productivity. In extremely stressful situations, employees and dog-owning managers said that the animals are more of a burden, e.g., due to the necessity to go for a walk, when the workload does not allow it during lunch. 

### 5.1. Flexible Organizational Culture and Policies

During the interviews and the analysis, it became clear that, in all companies, certain prerequisites are necessary to create an environment in which dogs can be at the office without being a burden on the owner. These findings were not revealed through questioning about requirements, but were instead mentioned by managers, as well as employees, when asked about their or the employee’s daily work behavior, dealing with problems, and if any restrictions are in place regarding dogs at the workplace. This is an indicator that these factors are not only considered relevant by management but have a real impact on the effect and even the feasibility of having pets at the office.

#### 5.1.1. Trial-And-Error Mentality

None of the managers imposed any restrictions regarding the number of dogs at the workplace. Statements regarding that topic, such as “*no, in the first line the dog is welcome and then let’s see how it works*”, suggest that an environment is created where things will be tried to see what works and where intervention is needed. The same flexibility was found when discussing breeds. This also gives companies a certain amount of flexibility when reacting to employees’ demands. While one manager stated “*the difficulty is that once it was allowed and then gradually came, you can’t forbid it to the others then*”. By having a trial-and-error mentality, the managers are able to intervene and create rules for action, if the workplace is disturbed.

#### 5.1.2. Open Communication

When problems occur regarding the dogs, all managers say that addressing the problem openly and directly is essential to make pet-friendly policies work. Employees observed that it is also important that managers take actions when dogs show disruptive behaviors. Problems cannot be ignored when the general work climate is compromised. Weekly face-to-face employee–supervisor meetings are often used to address such points. However, a certain degree of flexibility also reduces stress for the owner. When a manager is a dog-owner, the perceived pressure for dog-owners during or after incidents like barking or “accidents” decreases.

#### 5.1.3. Autonomy in Job Design

When asked how the dog at work changes the daily work behavior, most interviewees mentioned the need for breaks for the sake of the dog. Most employees said that their lunch breaks increase with the dog present, but it was also stated that, due to flexible hours and autonomy, this was not an issue. Looking at this from a different angle, inflexible work schedules would lead to greater pressure and increased stress for the dog-owner, possibly canceling out the benefits that having dogs brings. In addition, most managers mentioned that bringing a companion does take away some time, even during working hours: “*Surely the dogs partly rob a little working time. It’s got to be said*”. With that, flexible hours are crucial so that the animal does not have a negative impact on productivity.

### 5.2. Positive Influence on Job Satisfaction and Climate

All companies had the prerequisites mentioned above, and, in general, the interviewees believe that the company benefits from having the animals in the office. Four main areas were identified regarding how dogs can have a positive influence on job satisfaction, as well as organizational climate. Interestingly, those factors are not limited to the owners, but affect the whole company.

#### 5.2.1. Positive Work Environment and Stress Release

As the number-one reason for dogs at workplaces, interviewees reported a high impact of dogs on perceived stress and positive work environment. According to self-reports, the dogs help employees cope with stress. Short breaks to interact with the dog help to recharge and increase the mood, as shown in one employee’s quote: 


*So I somehow sit and hack on my laptop and talk on the phone and he comes and then I notice immediately, I take my time, look down at him, cuddle him, take him on my lap, then cuddle him again, and then I recharge my batteries in that moment, which maybe half an hour of break wouldn’t have brought me.*


However, it is not only the owners who profit from those short mental breaks; in most companies, it was reported that non-dog-owner co-workers play with the dogs as well, using this time as a mental break from their work.

This connection goes so far that co-workers get attached to the dogs. Another topic that has not yet received scholarly attention is the benefit of the mandatory lunch break for dog-owners. According to both managers and employees, going out for lunch with the dog is an enormous stress release, due to the fresh air and the exercise but also due to the simple fact of taking a break. Especially in high-workload companies, like marketing agencies, many workers skip their breaks in order to continue working. 


*They lead in any case to the fact that at least the dog-owners have to really do a lunch break, that can happen in agencies quite often that one tends to leave out the break if there is too much work at times. […] That was also for me personally one of the motives for having a dog.*


Another manager remarked on the influence of a walk for the employee’s work behavior:


*I honestly notice that the dog helps her because she goes out with him twice, she goes out twice and that’s sometimes very good and when she comes back with him after 20 min I get the impression that a few knots have loosened in the 20 min and it seems to be the fresh air.*


Lastly, the dogs have a positive impact on the work environment, bringing “*smile[s] to the other employees*” and “*sending out positive vibes*” for owners, co-workers, and management.

#### 5.2.2. Communication Improvement

Our data showed that dogs often act as an icebreaker. Employees and managers reported that dogs break barriers with customers and give a topic of conversation and bonding between customer and service provider. An additional effect is the positive impact on integration of new employees who own a dog. According to one employee, the dog helped her get in contact with the other dog-owners faster, to set up arrangements regarding the dog. This supports the hypothesis of Hall and Mills [38] that dogs facilitate social integration in a work setting. While the literature shows an increase in social interaction, the reasons were only guessed at. One of our managers (at German Shepherd) described a phenomenon that gives an insight of how dogs influence the communication inside the workplace even further:


*[D]ogs definitely contribute to the exchange across the teams because the dogs are such a connecting element. […] so even if I’m only with a team because my dog is running there, or vice versa, that’s just a lot more social exchange in the whole agency.*


Therefore, the existence of the dogs in the office can lead to a greater information exchange between different departments or supervisor and employees. Other interviewees also showed that dog-owners sometimes followed their dogs in other departments or that people came to visit the dog, which also increased social exchange.

#### 5.2.3. Social Cohesion

Dogs seem to not only foster social interaction and integration but also social cohesion. Dog-owners often support each other, creating a sub-community inside the company. The services range from going out with the dog for lunch when another owner has too much stress or an appointment, to even taking care of the dog when a colleague is on vacation or a business trip. This social cohesion can create a sense of belonging and commitment to the company and the colleagues.

This support also extends beyond dog-owners because non-owners sometimes take care of a dog when the owner is busy. This demonstrates that the positive influence of dogs on community [36] is replicated in the work setting as well, creating social support and cohesion, as well as a better-functioning community.

#### 5.2.4. Appreciation and Commitment

Employees express a high appreciation for the pet-friendly policies of their employers. They see it as privilege and as prestigious, leading to a favorable appreciation of the company. All employees sounded pleased and positive when speaking about their company’s pet-friendly policy. 

It also becomes clear, however, that restricting this freedom would diminish the appreciation of the company in the eyes of the dog-owners. Taking away this benefit in the short-term would lead to stress for the owner and a loss of motivation. In the long-term, most interviewees with dogs are certain that they would switch jobs if it would no longer be possible to bring their dog. 

Managers are aware of these consequences, and one even add that “*if we suddenly would say the dog is not allowed, I would probably lose two employees*”, referring to co-workers who got attached to the office dog as well. This observation supports the idea that dog-friendly policies are a tool to influence employee retention and acquisition. Many managers reported cases in which bringing the dog to work has been mentioned by the applicant as a requirement during job interviews, showing the importance to applicants. However, it also underscores the importance of consistency. First allowing and then later forbidding dogs in the office might lead to high turnover rates and therefore unwanted costs to replace the trained staff. 

### 5.3. Symbolic Functions of Dogs at Work: Brand and Values

#### 5.3.1. Applicant–Company Fit

When talking about applicants, managers were asked what impact it would have if an applicant did not like dogs or had an allergy. Managers see the dogs as part of the office’s environment, making it a factor that the applicant needed to take into consideration, not the managers. One manager even questioned the suitability of the applicant if the applicant would not want to work in an office with dogs present:


*[T]hen I would also immediately think, OK if he already starts like that, then he doesn’t fit in here either. That’s just the way it is in agencies, that everything is always very open and relaxed and that’s kind of like that.*


By including dogs on the websites as part as the “team”, or by including them in regular Facebook posts, potential applicants are also aware of the situation very early during the application process. In fact, one of the five companies often used its dogs for image and brand building by including the dogs on social media and the website. Two other firms use it sometimes, but do not focus on the dogs. For two companies, their company-controlled media do not show any office dogs.

#### 5.3.2. Values and Implications

Allowing dogs is not just a benefit for employees given by the company, but it also reflects on the company’s values regarding openness and flexibility. It is also a sign of how willing employers are to deal with employees’ needs and work–life balance. This becomes clear when looking at the quote of one manager: 


*So we want to give our employees as much freedom and entertainment as possible and work-life balance and life-life balance and opportunities, and for me that includes that [being able to bring the dog to work].*


Allowing dogs reflects openness and flexibility, but, on the other hand, it expresses a certain degree of employee focus from the supervisors. This influences employer branding, especially regarding the fact that most employer-rating sites include benefits such as pet-friendly policies in the filters.

### 5.4. Deriving a Grounded Theoretical Model

While Figure 4 shows the static data structure for the key themes that emerged from the interviews during the analysis, Figure 5 displays the dynamic processual relationships as a basis for the grounded theory model about the influence of pet-friendly policies on the organization and the necessary requirements regarding the organizational environment. 

With dogs adding a level of stress, the environment in a company has an impact on the extent to which dogs have a positive influence on the employees and the business itself. Dogs have a positive impact only if the company’s values include flexibility, open communication, and autonomy regarding working hours. An open and trial-and-error mentality of employees and managers creates a feeling of safety for dog-owners but also ensures that the work environment is not overly disturbed. Otherwise, having a dog at work that has its own rhythm could increase stress levels and decrease productivity for the owners. This is consistent with the general belief that it is highly important that the employee-benefit subsystem and policies are kept consistent with actual system designs and the culture and the company goals to have a positive impact [64]. If this is not the case, the success of strategies and policies are often limited [65].

Having a certain environmental context as a prerequisite for a working system also changes the impact of having pet-friendly policies in place on the image and reputation of a company. As stated by Cunha et al. [5] and Ferguson [50], pet-friendly policies can be used to project a certain image for the outside world as a company with a relaxed culture and informality. This study takes this assumption to another level. Pet-friendly policies are actually only possible when the company really is flexible and employees have autonomy regarding their work hours. This makes pet-friendly policies a far greater influence on the image and the employer branding as previously believed. This also raises the question of how to assess companies that have sought to adopt a pet-friendly policy and failed. 

Because we found out that social cohesion is a requisite to take pressure off the dog-owners and a reason they did not feel compromised in their flexibility when they brought the dog, a culture of community and support is also important. Very competitive environments with high individualism might hinder support between co-workers, making it unsuitable for pet-friendly policies. Because a company’s culture has an influence on peoples’ behaviors and problem-solving [66], it is also highly influential on the success of pet-friendly policies, as shown in this research. In general, accepting dogs in the office can also foster a more cooperative environment and better social support, and thereby create culture change. However, it needs to be kept in mind that cultural change can only be created when management behavior and communication are adopted as well [67,68]. Only then are the company and employees able to change the culture [69]. The study underscores that company strategy and culture need to be aligned with the human resource strategy and its measures and organizational requirements [70]. Therefore, not all companies can implement these policies. They need to fit the circumstances and the culture.

If these prerequisites are in place, however, pet-friendly policies foster job satisfaction and have a positive influence on the work climate. Small breaks to notice or play with the dogs are used by owners and co-workers, to release stress. Because stress negatively affects the well-being of employees and is related to burnout tendencies [71] and retention willingness [47], the reduced stress can improve company performance. Mandatory lunch breaks for dog-owners have a similar effect on productivity and well-being. In addition, greater social inclusion and social support (from co-workers, as well as the animal) increases well-being and job-satisfaction. Moreover, people seem to be happy about the presence of the dogs in the office and employees’ and supervisors’ communication increases when dogs are moving about freely in the office. At the company level, the dogs are a plus for dog-friendly workers and dog-affine clients but also increase social interaction and cross-department exchange, and thus increase the general social capital of a company.

Appreciation for the benefit of bringing their dog seems to influence commitment. However, while Hall and Mills [34] detected a lower willingness to leave the company, it is important to mention that, should the company take away the benefit of bringing the dog, owners might quit and look for a different job. 

This impact on climate and job satisfaction, as well as the existence of prerequisites, makes pet-friendly policies an important point in employer branding. Dogs at the office symbolize flexibility in work hours and policies and open communication. The positive influence of dogs on the organizational climate seems to be reflected in the general opinion of ex-employees. It represents a general people-first approach of the management, portraying that values like work–life balance are relevant for the company. It also means that potential applicants need to express a certain degree of flexibility as well. 

## 6. Discussion

Our results generally support the belief that dogs at the workplace can have a positive influence on individual and collective well-being of organizational members. The analysis shows that the positive influence of dogs on communities can also be transferred to the office environment, making this a new contribution to the study of dogs’ influence. Importantly, this is the first study linking the positive effects of dogs to certain prerequisites that need to be fulfilled in the company. Flexible hours and autonomy are key for the pet-owners to be able to deal with the responsibility of the animal during work. It is important to highlight that those requisites are not necessary in order to be able to implement pet-friendly policies, but they are highly relevant when companies seek to decrease stress for employees and improve the work atmosphere and social capital. 

Our findings have business implications. Although it becomes more and more common for “young and trendy” companies to allow dogs at work, companies should also consider if the culture is ready for it. Supposedly good incentives implemented in a wrong culture and framework might backfire and hurt the company by putting more pressure and stress on the owners, because, for example, breaks cannot be taken flexibly. 

While this research has showed the necessity of a certain flexibility and autonomy in a company to benefit from pet-friendly policies, this raises the question of what happens if this environment is not present. Further research should investigate what the extent of the consequences are for employees and the company if the culture and the management are not ready for dogs at the office. In addition, while the study showed that dogs enhance the social capital of a company, the degree of job satisfaction and performance is still underexplored. For this, investigating a company before and after implementing the policy might be of interest, because the prerequisites identified above are already associated with a positive climate, job satisfaction, and performance. A comparison between dog-less and dog-inclusive companies might be misleading.

The study has limitations, namely the fact that we studied five small organizations in the same sector and geography. This constitutes a limitation, as well as a boundary, condition to our study. Future research may explore how organizational characteristics, including age, size, and demographics, help to explain pet-friendly policies.

## 7. Conclusions

We focused on pet-friendly company policies and employees who bring their dogs to work. We concluded that participants are, in general and in principle, favorable to the presence of dogs in organizations. Our informants mentioned the positive impact on social cohesion, a feeling of community, and an increase of the information exchange. We showed that pet-friendly policies may be positive, but also that they are not always positive, revealing boundary conditions for organizational pet policies. Therefore, pet-friendly policies are only effective if they are implemented genuinely and hand in hand with the company’s culture and atmosphere and not as a mere instrument to increase job satisfaction and engagement or to improve employee branding.

## Figures and Tables

**Figure 1 animals-11-00089-f001:**
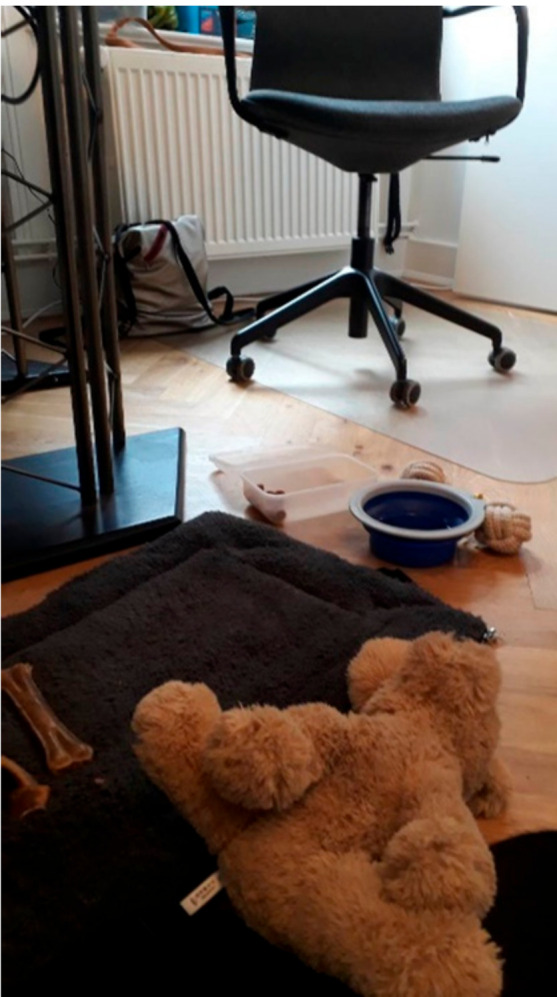
Dog’s place in the office.

**Figure 2 animals-11-00089-f002:**
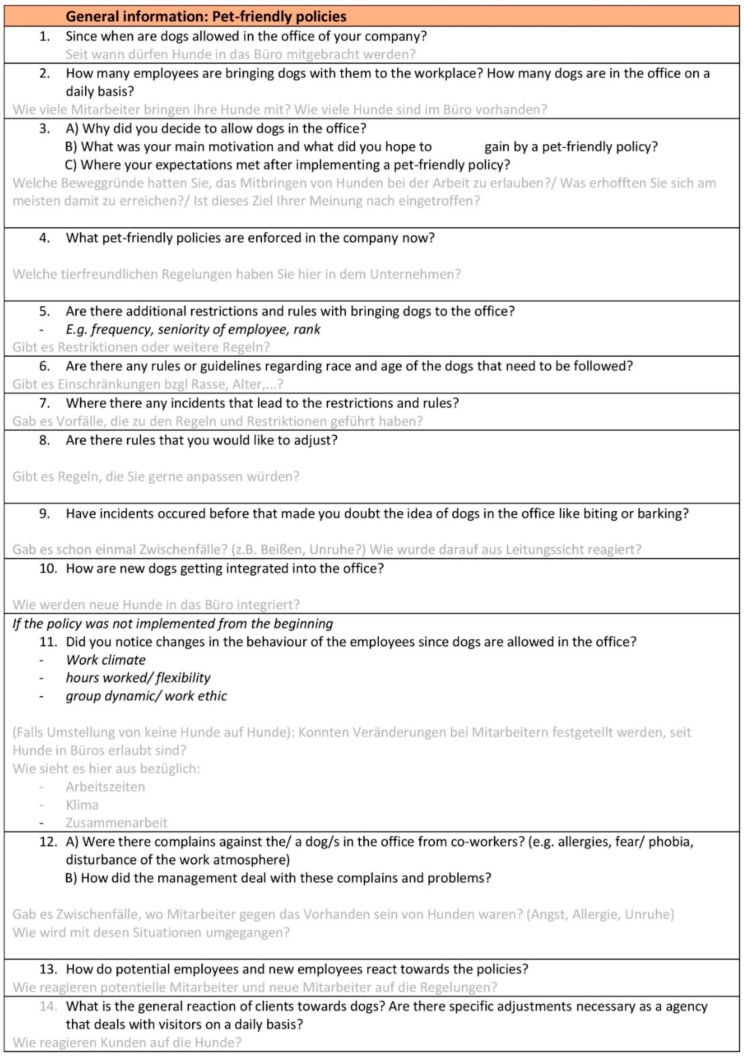
Interview guideline for manager.

**Figure 3 animals-11-00089-f003:**
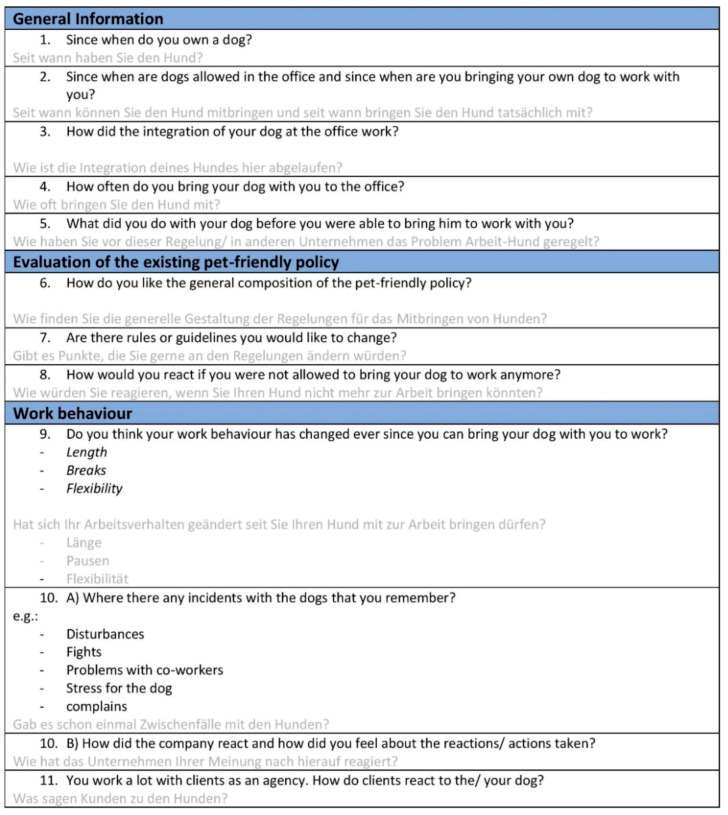
Interview guideline for employee who brings dog to work.

**Figure 4 animals-11-00089-f004:**
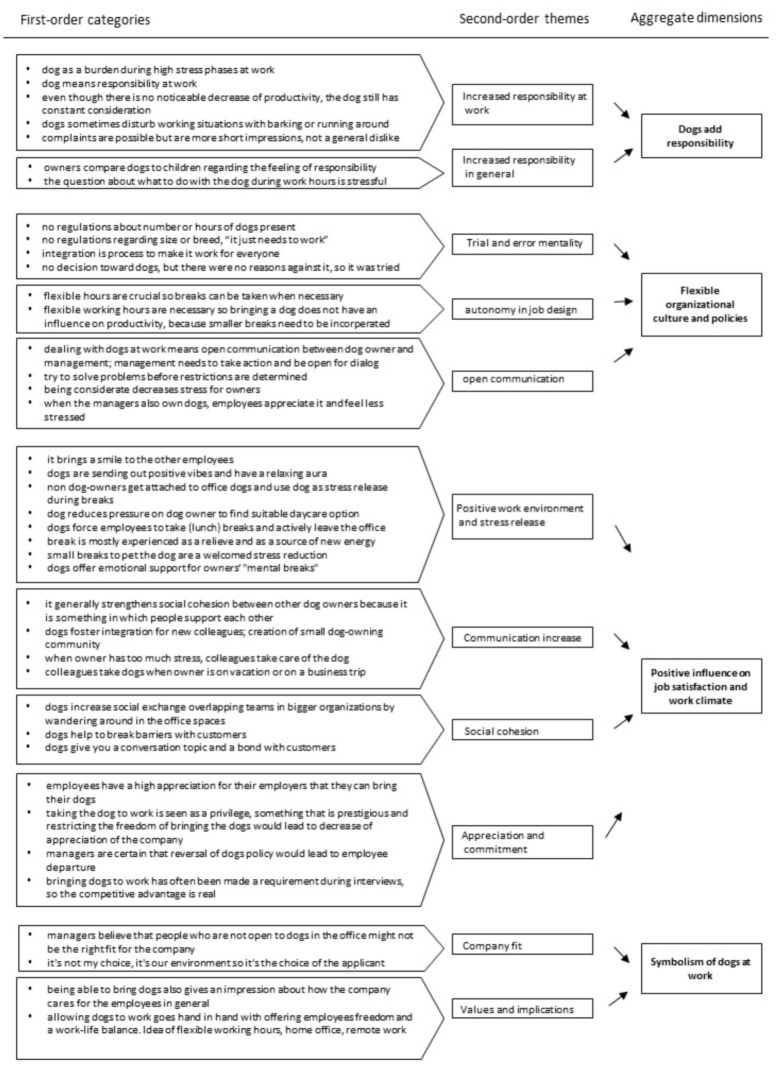
Data structure.

**Figure 5 animals-11-00089-f005:**
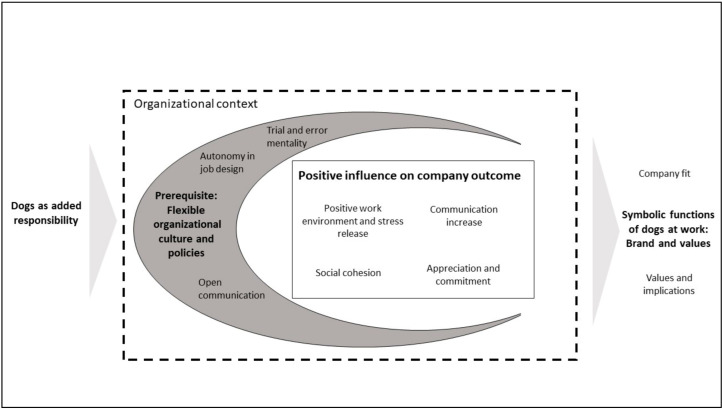
Grounded theory model.

**Table 1 animals-11-00089-t001:** Company overviews.

	Company Poodle	Company Bulldog	Company Labrador	Company German Shepherd	Company Beagle
Kind of company	Marketing agency	Film production	Marketing agency	Public Ralations Agency	Marketing agency
Number of employees	10	6	40	30	12
Number of dogs total	2	1	1–3	5	4
Number of employees bringing dogs	1	1	1–3	5	4
Time since dog-friendly policy	5 years	3 months	9 years	20 years	20 years
Number of interviews held	2	2	3	3	2

**Table 2 animals-11-00089-t002:** Quotes leading to first-order categories.

*“Going to the office without a dog is really more relaxed, because I also say, now I can have a date with a colleague, now I don’t have to go for a walk, now I can have dinner or something. Well, it is, it is a liberation sometimes”.* *“Well, I sometimes find it stressful to have a dog at work because I have this responsibility”.* *“If I’m so busy, I still have to go out”.* *“Bringing dogs with you of course also means obligation. Of course, you always have an eye on what they’re doing”,* *“There are sometimes situations where they bark. But that’s relatively short. So, if this were to degenerate into barking through permanently”.*	Increased responsibility at work
[Answer on the question what it would mean for the dog-owner if the dog could not come to work anymore] *“I would have to look what I can do with him the three days a week, so who takes care of him. It’s like a kid suddenly who doesn’t have a day care place anymore”* *“Because if you can’t bring your dog to work anymore, then you have a really big problem”.*	Increased responsibility in general
*“So there is no regulation, there is only a gut feeling and I would discuss it with all employees”.* *“So if you find out[that it doesn’t work] after the dog had a chance, after a long enough period of time, after you got to know the dog”* *“I would rather say there were no motives not to allow it”.*	Trial and error mentality
[Interviewee talking about breaks when the dog needs one] *“Which is not a problem, no one’s looking at the clock and says that’s not OK. I wouldn’t say now that, somehow, the productivity suffers. Well, no more as if there are people standing at the coffee machine for too long, so it’s not like we have a fixed time anyway, but you have to do what has to be done and how long that takes…so if you can do the work in 5 h, you can go home and if you need 10 h, then you need 10 h”.*	Autonomy in job design
*“[W]e have regular personnel sprints, i.e., four-eye interviews with the management and employees, and that’s where we talk about something like that [dog behavior]. Both positive and negative”.* *“So there was a time when a dog actually growled at this typical example of postman or something. But there was the solution that the colleague really brought a dog psychologist with her”.* *“[W]e have an agreement with all dog-owners that we also address critical situations, which can also lead to us prohibiting the bringing of the specific dog”.*	Open communication
- *“[I]t’s always like this when they are just walking around and sniffing at people and greeting that it’s always positive, so that it sends out positive vibes”* - *“[W]hen a dog is in the office, the mood rises”*. - *“[T]he two smaller ones are cooler too, bringing back or throwing balls. That’s something that the colleagues do more often”* - *“I honestly notice that the dog helps her because she goes out with him twice, she goes out twice and that’s sometimes very good and when she comes back with him after 20 min I get the impression that a few knots have loosened in the 20 min and it seems to be the fresh air”*. - *“[T]hat sometimes I can work even better through him because I get mental breaks or have to go out at noon and thus create new energy again”*.	Positive work environment and stress release
- *“It generally strengthens social cohesion because it is also something where people support each other”*. - *“[A]nd there you can see we also support each other a lot. We already have this small community, of course, because you can better assess what is missing among each other”*. - *“It actually goes so far when we have business trips or even holidays, we each other also take over the dogs”*. - *“[B]ecause then we laughed a lot and then the dogs found each other and therefore it was relatively easy for the integration for me as well”*.	Social cohesion
- *“Also, you just talk about them, this is often also a topic to talk about, both with customers and with service providers”*. - *“[W]ell it is actually always kind of nice when you have the little dog in advance sitting on your lap and then you already have a small topic to start the conversation right away”*. - *“[S]o even if I’m only with a team because my dog is running there, or vice versa, that’s just a lot more social exchange in the whole agency”*. - *“[T]he dogs definitely contribute to the exchange across the teams because the dogs are such a connecting element”*.	Communication increase
- *“But I’m thankful that it works that way now in this job at least”*. [Interviewee after being ask what would happen if he could not bring his dog to work anymore] *“and I would not think very well about the company any longer like I do right now”.* *“Has that ever come up during interviews?”* - *“Yes absolutely, that has definitely an influence for a lot of colleagues”*. [Interviewee after being ask what would happen if he could not bring his dog to work anymore] *“That would, yes, that would definitely lead to demotivation”.* [Interviewee after being ask what would happen if he could not bring his dog to work anymore] *“I have a dog and I can never see him and I would ask myself if that is really the place where I would have to work”.* *“I actually asked when I had my interview if I could take my dog with me”.*	Appreciation and commitment
- *“So we make sure that. at the latest during the at job interviews the applicant finds out that there are dogs here”*. *“I would say that the applicant, that we are no longer interesting for the applicant. And so, it is not so much my decision, but actually their decision, because that is our environment”.* *“So if someone here would say “Ah, do you really have dogs here, I can’t bear that at all”, then I would also immediately think, OK if he also starts like that, then he doesn’t fit in here either”.*	Company fit
*“[B]ut it is also a factor for every dog-owner, how my employer deals with the fact that I have a dog and because as an owner one has a responsibility and must plan continuously”.* *“So if something is important for an employee, then I do”.* *“So we want to give our employees as much freedom and entertainment as possible and work-life balance and life-life balance and opportunities, and for me that[being able to bring the dog to work] includes that”.*	Values and implications

## Data Availability

Data will available from the first author upon request.

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
