# Peer review of "Dogs at the Workplace: A Multiple Case Study"

_animals, 2021, doi:10.3390/ani11010089_

Round 1

Reviewer 1 Report

This is an important paper that I think will generate a lot of interest and further research.  The empirical research and literature base are sound. 

The argument that pet-friendly workplace policies are important but need to be genuine rather than instrumental is significant and important for the future welfare and ethics of companion species, and not just humans and their organisational workplaces.

Minor points:

The abstract should give more clear methods if there is room in the word count – so explain more specifically what you mean by ‘empirically scrutinize’.

Para 2:  What is meant by ‘organisational members’?

Para 3:  data is quantitative and qualitative – not the study.  Can you express this differently … e.g. ‘based on qualitative data’?  Also perhaps express ‘zooming in’ differently as at first I wasn’t sure if the study was done via “Zoom” (given the COVID-19 pandemic).   Perhaps ‘analysing’ or ‘interrogating’ interpretations if that is what is meant?

Page 2, para 2:   why is happiness subjective?  If participants are reporting this it is their experience so they are the best indicator of their feelings?  I’d suggest deleting subjective.  Similarly the next paragraph talks about self-evaluations as subjective but I think this is a tautology and unnecessary.   It also privileges measurable data over other forms of knowing which I don’t agree with.   It may be part of another discipline e.g. in the human behavioural sciences which I’m not familiar with but it seems strange and arguable to a social scientist.

Page 3, para 3:  what is meant by work ‘engagement’, is it how focussed someone is on the work?  Or the way they ‘engage’ with others?  Is there a better word that is more precise?

Page 5, para 4:  what archival data were used in conjunction with the interviews?

Page 5, last para:  this para on the questions (method) should go before the analysis – the development of the categories, or else into the next section on data analysis I think?  Also the ‘basic interview guide’ is unclear – is this a semi-structured interview where some questions are developed in advance and then the answers of interviewees are probed further with unknown questions to elicit further meaning from their answers?  I think this is what the authors are saying but if not, what type of interviews?

Page 6, second para:  Is there a year to follow Gioia?  Or else explain what Gioia is.

Findings

This section has mostly good quotes, some of the interpretation is a little repetitive and perhaps could be reduced (which may also require removal of a few quotes).   I realise there is a structure followed in presenting the evidence, but I think this section could be made slightly ‘punchier’, particularly given the figures and tables that can assist the reader with the detail.

Deriving a grounded theoretical model:

This section tries to balance the argument but I think some of the ‘negatives’ of pet-friendly organisations are easily countered.  The pet-friendly policies need to be matched with policies about pet-owner behaviour.  Although findings reveal that organisations allow dogs to be at the office where they are not a burden on the owner (page 7), isn’t there also a need for the owner to be a responsible pet owner?   That would then alleviate some of the problems with barking at or stealing food from other dogs – reported as negatives?  Such responsible pet ownership also then alleviates the problems of managers not wanting to look like they are allowing some dogs but not others, also reported in the findings.   Are these part of the rules that can be created by managers?

Also in terms of the findings, and although I agree with dogs having a comfort break, the same dogs possibly wouldn’t have the comfort break, were those dogs left at the owners home alone during the day.  So comfort breaks for the dogs surely wouldn’t be required much and could be part of the usual employee breaks (e.g. lunch break, coffee break).  In that way, all dogs would normally be comfortable throughout the day, having been walked, toileted etc as per the usual routine of dogs not allowed into workplaces and only in rare circumstances would they need an additional break. 

The collectives of people and socialised dogs so that co-workers could give each other’s dogs a break when a particular employee was not able to toilet their own dog is interesting.  There findings are important and I think easily also worked into responsible pet owner behaviour codes/rules by managers.  I realise this is a different paper, but I think the discussion could allude to this potential to create such pet-friendly policies (alongside responsible pet-owner policies mentioned above) and then perhaps that becomes a platform for further research. 

Another potential benefit is the benefit for clients.  I have walked into pet-friendly workplaces and immediately felt at ease by the staff and accompanying dog – the benefit to the company from satisfied, happy clients and the atmosphere it creates is also potentially an area of further research.

Discussion

This section seems largely repetitive of the previous section on deriving the theoretical model.  I’d suggest deleting some of this and integrating the two sections, and then perhaps looking into a stronger discussion that brings in more counter-arguments. 

Conclusion

This section should, whilst summarising each section, tie up the key points from the discussion and re-state the argument as it isn’t clear what the authors are arguing.    So the Discussion/Theoretical model sections should point to the argument mentioned in the abstract that pet-friendly workplace policies are important but need to be genuine rather than instrumental.  Explain what is instrumental versus what is ‘genuine’ along the way so you can then make this claim.

(Also delete the last paragraph which seem to be instructions.)

General:

The work needs a final editorial polish.

Author Response

Dear Reviewer,

Thank you for your comments and suggestions. We responded carefully and  explained how the paper was changed in the attached document. In case you consider that more is necessary, please let is know. Thank you, Elisa and Miguel

Reviewer 2 Report

This paper describes a qualitative study exploring the effects of dogs in the workplace on dog-owner and non-owner employees and managers. This is an interesting topic which hasn't been studied enough. How relevant it will be post-COVID remains to be seen, with many people simply working from home nowadays. Nonetheless, the idea is worthy of publication. The ms in its current form needs some work before I can recommend it for publication, though. My main concerns are with the abstract, the first subsection of the intro, and some of the descriptive quotes used to illustrate the themes.

The abstract is very limited. The authors should keep in mind that many readers will only read the abstract, so it needs to be sufficient to tell the story of the ms as a standalone. There is no introductory sentence to explain why this topic matters. There is also no concluding sentence with implications. What IS there isn't always clear. It's not evident what 'trial and error' mentality means until one reads the full text. The abstract should be expanded upon and clarified.

L18 add a citation after 'provided some debate'

L29 the sentence about evolving data analysis is unclear.

L33 genuine vs instrumental approach is unclear.

Section 2.1. is weak relative to the rest of the intro. The authors could consider shortening it just to give a brief overview before launching into the dogs in the workplace section. A paragraph would be sufficient to provide a general overview. If the authors decide to keep it as long as it is, I suggest the following:

L41-42 cite claim about various studies. This para should also mention that plenty of studies have shown no effect, or even a negative effect. These are mentioned later but should also be here.

L43 cite claim that 'the general belief holds...' and be more circumspect about the mixed results.

L43-46 AAT is different from pet ownership and from casual interactions with an animal. This doesn't fit well here.

L66-L71 is a good analysis.

L79-83 is a generally good analysis, but 'non-consensual' doesn't make sense here. Also, Parslow and Jorm's paper showed a NEGATIVE effect, not lack of relationship as implied here. They also found that even the additional mild physical activity associated with dog ownership did not lead to any improved cardio outcomes.

L93 suggest removing 'specifically' from header title

L132-133 cite claim about managers noticing the influence of dogs

Overall this was a good subsection.

L163 in addition to the dislikes/allergies/phobias, mention that dangers and distractions can also be an issue, to set the reader up for the whole subsection, rather than just the first portion.

L165 italicise subsection name

L179 cite claim that dog bites are rare

L212 why was the creative sector chosen for this work?

Suggest moving Table 1 just below para ending on L214, where Table 1 is first referred to in text.

P222 suggest starting a new para with sentence 'interviews are an efficient way...'

L247 suggest adding the interview schedule as Supplementary Material, for ease of replication by other groups

When was data collected? Since we're now living through COVID, this info is more important than ever

L251 'place of dog' is unclear

L254 how were websites and social media analysed?

Table 2 would be better in the results section. I suggest not referring to Fig 2 in text either, until just before it is to be presented.

L275 phenomena is plural. It should be phenomenon, the singular.

L285-288 goes in results

L297 it was said by whom? employees or managers? or both?

L363 why insert Fig 2 so late? It should be near the top of the results.

Many of the illustrative quotes are unclear because they are not provided with sufficient context. It may be necessary for the authors to add info in brackets to add relevant context. This includes quotes starting on lines: 409, 428, 449

L418 italicise section header

L444 I think this is not the best header title for this section. It is unclear. Maybe something better would be something about how the dog policy reflects the company's identity. 'symbolism of dogs at work' is confusing

L461, 466, 468 - italicise

L586 - remove para

Fig 2 the quotes in 'autonomy in job design' seem almost identical.

Fig 2  why is the last point in open communication included under this theme? It doesn't seem related

Table 1 many of the quotes are unclear. These are: last quote in row 1; first quote in row 2; middle quote row 3; first quote row 4; first quote row 5; second quote row 6; fourth quote in social cohesion; second, fourth, and fifth quote in appreciation; last two quotes in values. for the first quote in company fit, 'at the latest' should probably be 'at the least'.

Author Response

Dear Reviewer,

Thank you for your comments and suggestions. We responded carefully and
explain how the paper was changed in the document attache. In case you consider that more is necessary, please let us know. Thank you, Elisa and Miguel
